

# Predatory blue crabs induce stronger nonconsumptive effects in eastern oysters *Crassostrea virginica* than scavenging blue crabs

Avery E. Scherer, Miranda M. Garcia and Delbert L. Smee

Department of Life Sciences, Texas A&M University-Corpus Christi, Corpus Christi, TX, USA

## ABSTRACT

By influencing critical prey traits such as foraging or habitat selection, predators can affect entire ecosystems, but the nature of cues that trigger prey reactions to predators are not well understood. Predators may scavenge to supplement their energetic needs and scavenging frequency may vary among individuals within a species due to preferences and prey availability. Yet prey reactions to consumers that are primarily scavengers versus those that are active foragers have not been investigated, even though variation in prey reactions to scavengers or predators might influence cascading nonconsumptive effects in food webs. Oysters *Crassostrea virginica* react to crab predators by growing stronger shells. We exposed oysters to exudates from crabs fed live oysters or fed aged oyster tissue to simulate scavenging, and to controls without crab cues. Oysters grew stronger shells when exposed to either crab exudate, but their shells were significantly stronger when crabs were fed live oysters. The stronger response to predators than scavengers could be due to inherent differences in diet cues representative of reduced risk in the presence of scavengers or to degradation of conspecific alarm cues in aged treatments, which may mask risk from potential predators subsisting by scavenging.

## INTRODUCTION

Predator deterrence is a necessary but costly process for prey (*Baldwin, 1998*; *Weissburg, Smee & Ferner, 2014*). To minimize costs, prey utilize plastic defenses which are induced in situations of high predation risk. These defenses include changes in behavior, morphology, life history, or a combination of responses (*Kats & Dill, 1998*). For example, many organisms reduce feeding activity (*Smee & Weissburg, 2006*; *Large, Smee & Trussell, 2011*), utilize refuge habitats (*Mirza & Chivers, 2001*; *Schoeppner & Relyea, 2005*), speed up reproduction (*Laurila, Kujasalo & Ranta, 1998*; *Kiesecker et al., 2002*), delay reproduction (*Covich & Crowl, 1990*; *Laurila, Kujasalo & Ranta, 1998*), and produce structural (*Harvell, 1986*) or chemical defenses (*Pawlik et al., 1995*; *Baldwin, 1998*) under conditions of risk. Through these induced defenses, predators produce nonconsumptive effects which can have significant consequences for prey that may influence entire

Corresponding author
Avery E. Scherer,
avery.scherer@tamucc.edu

ecosystems in ways that are equal to or stronger than the effects of direct prey consumption (*Preisser, Bolnick & Benard, 2005*). However, although these induced defenses often increase survival, organisms employing them typically incur reductions in growth and/or fecundity. As a result, there is considerable pressure on prey to accurately assess predation risk and respond accordingly to balance energy intake and investment in growth and reproduction with the need to avoid and deter predators (*Harvell, 1990*; *Ferrari, Wisenden & Chivers, 2010*). Thus, understanding factors that influence the induction of prey responses can improve our understanding of nonconsumptive predator effects (*Weissburg, Smee & Ferner, 2014*; *Scherer & Smee, 2016*).

Prey rely on risk cues, often predator exudates, to accurately assess and react to predation risk (*Ferrari, Wisenden & Chivers, 2010*). Predator exudates may provide a variety of information including predator species and density that articulate the severity of the threat posed (*Weissburg, Smee & Ferner, 2014*). Chemical cues may also vary with a predator's diet and, as predators may change diets seasonally or with environmental conditions that affect foraging success (*Huggard, 1993*; *Chivers & Mirza, 2001*), and prey can incorporate information from predator diets to ascertain relative risk and react appropriately. Predator diet cues can reflect hunger state, prey species preferences, or larger scale dietary classifications (i.e., carnivore versus herbivore, generalist versus specialist) (*Scherer & Smee, 2016*). For example, carnivores release sulfurous compounds after digesting meat, and these compounds trigger risk responses among a variety of prey species that do not occur when carnivores are fed herbivorous diets (*Nolte et al., 1994*). Likewise, juvenile fish avoid habitats with predators fed a piscivorous diet as opposed to invertebrates, even though those fish are not naturally piscivorous (*Dixson, Pratchett & Munday, 2012*). Studies on cues that reflect the dietary classification of predators can provide insights into the types of cues that prey use to evaluate risk and into mechanisms that drive broad ecological patterns such as habitat selection and the propagation of nonconsumptive predator effects in food webs.

Yet studies of how larger scale dietary shifts beyond changes in the prey species consumed influence prey response and cascading nonconsumptive effects are rare (*Scherer & Smee, 2016*, but see *Nolte et al., 1994*; *Dixson, Pratchett & Munday, 2012*). For example, predators often supplement their energetic needs by scavenging and, although evidence suggests scavenging is widespread (*DeVault, Rhodes & Shivik, 2003*; *Wilson & Wolkovich, 2011*) and can affect communities (*Huggard, 1993*; *Barton et al., 2013*; *van Dijk et al., 2008*), both obligate and more often facultative scavenging are greatly underestimated in food webs (reviewed by *Wilson & Wolkovich, 2011*). Many generalist predator species are known to show foraging preferences for particular prey at an individual level (*Toscano et al., 2016*) and predators may show similar preferences for foraging or scavenging depending upon prey and carrion availability, competition, predator density, and physiological state (*Barton et al., 2013*; *DeVault, Rhodes & Shivik, 2003*; *van Dijk et al., 2008*; *Wilson & Wolkovich, 2011*; *Mattisson et al., 2016*). Although active predators and scavengers might consume similar prey items, diet cues may reflect these differences if cues change with time. As predators pose a greater threat to prey than scavengers, they should induce more intense defensive responses. However, studies comparing prey
reactions to individuals that scavenge versus active predators have not been performed, and it is unknown if prey can distinguish between predators and scavengers.

We used eastern oysters *Crassotrea virginica* and blue crabs *Callinectes sapidus*, which are oyster predators and opportunistic scavengers, as a model system for studying how predator foraging preferences influence prey responses. Oysters are an ecologically and economically important species (*Grabowski & Peterson, 2007*), and oysters are known to produce heavier and stronger shells under conditions of predation risk that effectively reduce mortality by crabs (*Robinson et al., 2014*; *Scherer et al., 2016*). Blue crabs are opportunistic predators which will readily scavenge dead tissue. Thus, we were able to feed the same predator tissue from the same prey species to avoid confounding results with species effects, and differences in oyster responses observed are solely attributable to differences in the age of consumed tissue. We expected oysters would respond to both predator and scavenger treatments, but that responses would be weaker when predators were fed aged tissue simulating scavenging.

## METHODS

Oysters (initial length 4.9 ± 0.13 mm) were purchased from the Auburn University Shellfish Laboratory Alabama, USA. These oysters were spawned from brood stock and reared for 1.5 months in common conditions before shipping. Oysters were maintained at Texas A&M University-Corpus Christi, Corpus Christi for 4 months in 37 semi-transparent static tanks (1050 mL, $n = 13$ tanks for control, $n = 12$ tanks per predation cue treatment) containing artificial seawater (Instant Ocean®, Blacksburg, VA, USA) prepared at a salinity of 20 ppt. Each tank contained 20 oysters and was randomly assigned to one of three treatments: control (no-predator exudates), active predator (exudates from crabs fed fresh oyster tissue), and scavenger (exudates from crabs fed aged oyster tissue). Water in the oyster tanks was replaced weekly, after which 350 mL of treatment (containing crab exudates) or control water (identical to oyster tank water) was added. Oysters were fed 5 mL of Phytoplex™; ENT Marine, Franklin, WI, USA phytoplankton per tank every other day and the laboratory was maintained at a temperature of 74 °F.

Blue crabs (carapace width 10–13 cm, mean 10.7 cm) were collected from estuaries near Corpus Christi, TX, USA. Eight crabs were individually housed in 20 L aquaria containing artificial seawater and maintained on one of two diet treatments (four crabs per diet). Predator treatments received 0.8 g of freshly shucked adult oyster tissue per week. Scavenger treatments received 0.8 g of adult oyster tissue that was shucked and weighed, then refrigerated for 96 h prior to feeding. Oyster tissue can degrade quickly in natural conditions and can be difficult to collect and to weigh. Therefore, we elected to refrigerate the tissue so that it could age but would not decompose to a point beyond use for our purposes, although refrigeration probably led to a conservative estimate of scavenger effects. The 96-h time frame was consistent with earlier studies (*Scherer et al., 2016*). All oysters used to feed crabs were collected locally.

Water changes were conducted weekly to maintain oyster and crab health and to prevent cue build up. Each week, following established protocol, treatment water was collected immediately prior to crab feeding. This allowed a week for cues to build up in
crab tanks. Crabs were then allowed ~1 h to eat, following which water changes occurred on all tanks. Performing water changes following crab feeding removed any excess oyster tissue, ensuring that crab cue treatments contained no prey material (i.e., contained only cues from the crab predators).

After four months of cue exposure, 10 oysters were randomly selected for analysis. All 10 oysters were measured to the nearest 0.01 mm for length and width using manual calipers. Length was measured as a distance from the umbo to the shell edge opposite initial shell growth. Width was measured at the widest point on the shell perpendicular to length. This data was then used to calculate shell surface area using the formula for an ellipse ($A = \pi r_1 r_2$). Due to the destructive techniques for measurement, we were unable to measure both weight and strength on the same individuals. Therefore, oyster shell weight was measured in five oysters and crushing force measured in an additional five oysters from each tank. To measure shell weight, soft tissue was removed and shells were dried at 38 °C for 48 h, and weighed to the nearest 0.001 g. Crushing force was measured using a Kistler 5995 charge amplifier and Kistler 9203 force sensor. This probe has a high sensitivity to measure small changes in force; the settings for the charge amplifier were sensitivity 47.8 N and range 50 N. A small blunt probe (1 mm diameter) was consistently placed centrally to be equidistant from the shell's edges and perpendicular to the surface for all oyster spat tested. We applied gentle and consistent pressure on all specimens until structural failure of the oyster's shell occurred (sensu *Robinson et al., 2014*). All data are available in both *.csv and *.jmp format in Supplemental Information.

### Data analysis

All analysis was completed in JMP Pro 12. For all shell metrics (shell surface area, strength, and weight), we calculated a mean value by tank, and used the tank value as our unit of replication. Shell weight and strength were compared among treatments using ANCOVA. Risk cue treatment was a fixed factor in the models and shell surface area was treated as a covariate to incorporate potential effects of oyster size on shell characteristics. ANCOVAs were first performed by including interaction terms between covariates and treatments to test for significant interactions. No significant interactions between treatments and covariates were found, indicating our data met the assumption for ANCOVA of homogeneity of regression slopes and allowing us to remove the interaction term from the model. Linear contrasts were used for comparisons between predation risk and control means and between fresh and aged predation risk means for significant ANCOVA results.

### RESULTS

For oyster shell strength (Fig. 1), the data fit the assumption of no interaction between predation risk treatment and shell length ($F_2 = 1.53$, $p = 0.23$). Treatment ($F_2 = 9.72$, $p = 0.001$) and surface area ($F_1 = 5.81$, $p = 0.02$) were found to be significant in the final model. Oysters increased shell strength to both predation cues ($F_{1,33} = 13.73$, $p < 0.001$), but strength was higher in active predator than in scavenger treatments ($F_{1,33} = 5.66$,

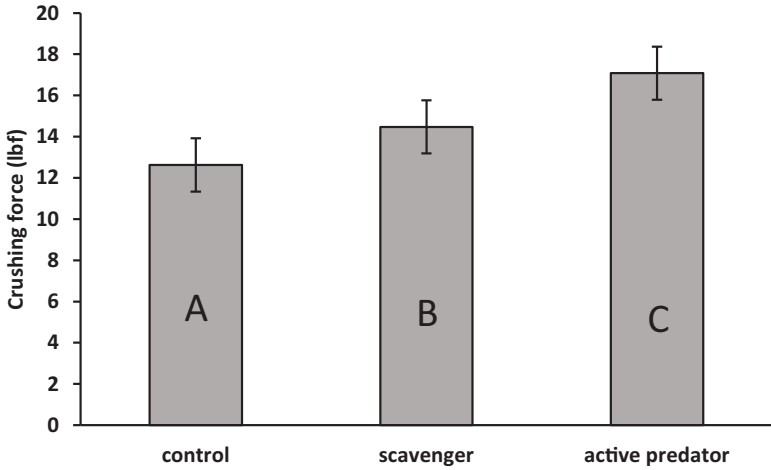

**Figure 1 Mean ± SEM shell crushing force.** Letter designations indicate significant differences at α = 0.05 as indicated by linear contrasts.

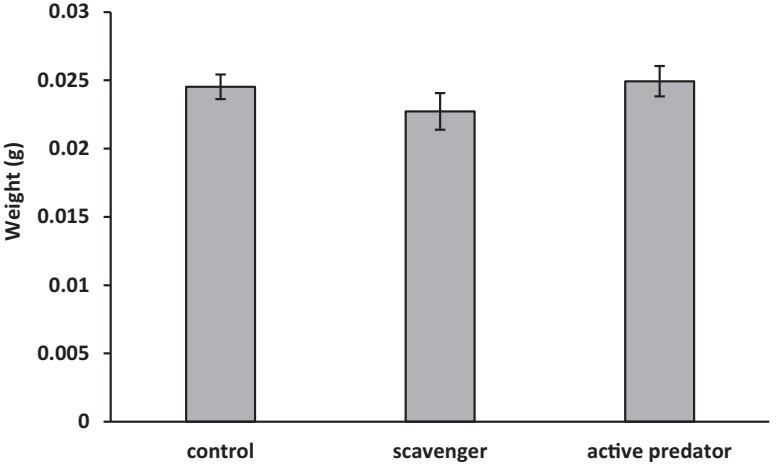

**Figure 2 Mean ± SEM shell weight.** No significant differences were found between predation treatments.

$p = 0.02$). For oyster shell weight (Fig. 2), the data fit the assumption of no interaction ($F_2 = 0.64$, $p = 0.53$). A significant effect was found for surface area ($F_1 = 6.48$, $p = 0.02$), but not for risk treatment ($F_2 = 1.03$, $p = 0.37$). Thus, oysters were stronger but not heavier in predator treatments compared to scavenger treatments and controls.

## DISCUSSION

The quantity and characteristics of available risk cues detected by prey organisms can influence the type and degree of reactions to risk (*Scherer & Smee, 2016*). Predator diet may influence prey responses, and predators consuming conspecific prey often trigger larger responses (*Schoeppner & Relyea, 2005*; *Weissburg, Smee & Ferner, 2014*). Prey may perceive other differences in consumer diets (carnivore versus herbivore) and our results indicate prey react differently to individuals of the same species engaged in active predation versus scavenging. Oysters grew stronger shells in response to crabs fed both

fresh and aged tissue as compared to controls, but were significantly stronger in treatments with crabs fed fresh oyster tissue. This finding is consistent with earlier work indicating oysters vary defenses in response to cues that reflect different risk levels (*Scherer et al., 2016*).

Adult blue crab claws (males, 100–165 mm carapace width) can exert a maximum force of $111.2 \pm 33.5$ N (*Blundon & Kennedy, 1982*). Although this is considerably greater than even the greatest forces needed to crush oysters in this study, changes of the magnitude seen here are likely to be biologically relevant for smaller crabs which are likely to target this size class of oysters. Further, blue crabs are known to practice size-selective foraging (*Ebersole & Kennedy, 1995*), which may make relative differences in shell strengths relevant, even when absolute strength values are far below what crabs are capable of overcoming.

Two potential mechanisms may explain different degrees of defense induction by oysters: differences in cue salience or in cue quantity. Under the first scenario, oysters respond to qualitative differences in cues which represent variation in the threat individual crabs pose to live prey. Oysters are known to react to alarm cues of injured oysters (*Scherer et al., 2016*) and may detect qualitative differences in the digested alarm cues released by predators indicating the age of consumed tissue. Additionally, individual crabs are known to show preferences for different prey based on defense induction and nutritional content (*Ebersole & Kennedy, 1995*). And growing evidence suggests many species considered to be generalist predators show individual foraging preferences due to trade-offs in foraging efficiency (*Melcer & Chiszar, 1989*; *Jackson & Li, 2004*; *Toscano et al., 2016*) which influence prey use of diet cues (*Pillay, Alexander & Lazenby, 2003*). Similarly, within consumer populations, some individuals scavenge more than others and scavenging rates can vary temporally with availability of carrion and physiological condition of the predator (*Barton et al., 2013*; *DeVault, Rhodes & Shivik, 2003*; *Wilson & Wolkovich, 2011*; *Mattisson et al., 2016*). Thus, differences in cue salience may represent ecologically valuable information about predator threat and prey may use diet cue characteristics to identify active predators that present greater risk.

Alternatively, the aging process may degrade cue components necessary for response induction. In this scenario, differences in oyster responses reflect a dearth of information, rather than differences in predation risk. Analogously, prey react less intensely to starved crabs (*Yamada, Navarrete & Needham, 1998*; *Griffiths & Richardson, 2006*; *Smee & Weissburg, 2006*; *Large & Smee, 2010*) despite presumably increased risk when predators are more motivated to forage (*Smee & Weissburg, 2006*; *Large & Smee, 2010*). This is perhaps because starved crabs release fewer exudates, as would scavengers if alarm cues are broken down before consumption. Reduced prey responses to starved crabs may increase crab foraging success if prey are less able to perceive them (*Smee & Weissburg, 2006*), and likewise scavenging may increase the likelihood of future successful predation events if scavengers are not perceived as risky.

We were surprised to find that the oysters did not change shell weight in response to predation cues. Previous studies have shown oysters consistently increase shell weight in response to blue crab predators (*Robinson et al., 2014*; *Scherer et al., 2016*). However, oyster

shells consist of two components: a protein matrix, which contributes heavily to shell strength but is expensive and slow to produce; and calcium carbonate, which contributes less to strength but is fast and cheap to produce (*Palmer, 1983*, *1992*; *Lee, Applebaum & Manahan, 2016*). Which of these materials is altered under conditions of risk is still unclear and there is evidence to suggest both may be involved (protein *Newell, Kennedy & Shaw, 2007*, calcium carbonate Scherer, 2014, unpublished data). There is also evidence to suggest factors, such as predator species, may influence how shells are altered (*Newell, Kennedy & Shaw, 2007*; *Robinson et al., 2014*). As this is the first such experiment investigating oyster defenses within this size range, it is unknown if size may also affect the mechanism of defense induction. Further research into this mechanism is needed and will contribute greatly to our understanding of these morphological defenses.

Inducible defenses initiate nonconsumptive predator effects that can significantly affect top-down forces and community structure (*Preisser, Bolnick & Benard, 2005*). But these effects are often context dependent, variable in time and space, and difficult to predict and model (*Weissburg, Smee & Ferner, 2014*). A more thorough understanding of factors influencing prey defenses could contribute strongly to our understanding of nonconsumptive effects. For example, the identity and nature of cues underlying prey reactions to predators remains largely unknown, hindering ecologists' ability to understand conditions that promote or inhibit propagation of nonconsumptive effects in food webs (*Weissburg, Smee & Ferner, 2014*, but see *Weissburg, Poulin & Kubanek, 2016*). Studies on dietary classifications (e.g., piscivorous fish versus fish preying on invertebrates, carnivore versus herbivore) may provide insights into the characteristics of risk cues accounting for broad ecological patterns. In particular, scavenging is often underappreciated and poorly quantified in natural communities (*Wilson & Wolkovich, 2011*), although switching from predation to scavenging can influence a suite of ecosystem processes (*Barton et al., 2013*). Here, we show that individual predators of the same species can elicit different prey responses based on whether they scavenge or consume live prey, indicating that scavenging rates likely influence the magnitude of nonconsumptive effects. By understanding the foraging preferences of predators and characteristics of cues that govern prey reactions to risk, ecologists can more accurately assess factors that govern the prevalence of nonconsumptive effects.

### Funding
This research was supported by NSF-MSP ETEAMS Grant #1321319 and by an NSF HRD LSAMP Grant #1304975. The funders had no role in study design, data collection and analysis, decision to publish, or preparation of the manuscript.

### Grant Disclosures
The following grant information was disclosed by the authors:
NSF-MSP ETEAMS Grant: #1321319.
NSF HRD LSAMP Grant: #1304975.

## Competing Interests

The authors declare that they have no competing interests.

## Author Contributions

- Avery E. Scherer conceived and designed the experiments, performed the experiments, analyzed the data, contributed reagents/materials/analysis tools, wrote the paper, prepared figures and/or tables, and reviewed drafts of the paper.
- Miranda M. Garcia conceived and designed the experiments, performed the experiments, analyzed the data, wrote the paper, prepared figures and/or tables, and reviewed drafts of the paper.
- Delbert L. Smee conceived and designed the experiments, analyzed the data, contributed reagents/materials/analysis tools, wrote the paper, prepared figures and/or tables, and reviewed drafts of the paper.

## Data Deposition

The raw data has been supplied as Supplemental Dataset Files.

## Supplemental Information

Supplemental information for this article can be found online at http://dx.doi.org/10.7717/peerj.3042#supplemental-information.

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
