# Peer review of "Predatory blue crabs induce stronger nonconsumptive effects in eastern oysters Crassostrea virginica than scavenging blue crabs"

_PeerJ, doi:10.7717/peerj.3042_

## Round 0.1 · original submission · Minor Revisions

I now have 3 reviews of your manuscript from experts in the subdiscipline, and all three reviewers were positive in their assessment of the article. All three reviews include comments designed to improve the ms. on revision; in particular, Reviewer 1 suggests that the writing be improved in the Introduction.

Please include point-by-point responses to reviewer comments in your rebuttal letter.

Reviewer 1 ·

Basic reporting

This study investigated how oysters respond to predator and scavenger cues created by blue crabs fed different quality of tissue. The authors found that exposure to cues of crabs that were fed oyster tissue had stronger shells than control oysters, but shell weight did not differ by treatment.

The Introduction includes relevant information about the subject with appropriate literature cited, but needs substantial restructuring. Multiple paragraphs begin on one topic and then do not provide appropriate support for that topic sentence. For example, in paragraph 2, the authors state that understanding dietary classifications of predators is important. However, they do not elaborate on why or how such classifications may differ, but expand on predator behavior (scavenging vs. foraging). This should be tied back in with how behavior may ultimately affect the diet and how diet might affect prey response to cues. The Introduction should also elaborate on what evidence there is for the 'plastic defenses' of prey referenced in in line 36. There is one example at the end of that paragraph, but such a statement needs more support from the literature. See other notes on structure of text in Discussion, line 127.

See note on line 148 of Discussion about nonconsumptive effects. This is a very rich topic that should be brought up sooner if going to couch results in this context.

Experimental design

There were no specific hypotheses/research questions stated, though it can be inferred from the information in lines 64-65 that they expected to see increased shell strength. However, it would be helpful to have a statement indicating how the authors thoughts those results would translate into their specific study question.

See in-text comments on description of methods. Overall, a very straightforward design and sufficient analyses.

Validity of the findings

See in-text comments on providing means and SE for oyster shell weight and inquiry on the lack of inclusion tissue weight.

Additional comments

Overall, I think this is a clean, straightforward study that provides important information towards understanding how predators can affect their prey.

Annotated reviews are not available for download in order to protect the identity of reviewers who chose to remain anonymous.

Reviewer 2 ·

Basic reporting

a. The paper is properly, professionally formatted and clearly written in English. The raw data and code for analysis are shared.

b. There are parts of the code that do not work because source codes are not attached, e.g., source("nlmeDiagPlots.R").

c. There are some small errors in the text:
• L72: give units for salinity
• L94: CVS should be *.csv
• L171, L181, etc.: need to italicize species names in references
• Crushing force data: for tank 13, replicate 5, crushing force = 0. Is this an error? Not sure what could give you a 0 here.

Experimental design

• The experiment is original primary search. It is a relatively simple study, however the methods section requires more detail, especially with regard to creating the three cue treatments.
a. L79: How many 20L aquaria did you use? How many crabs per aquarium?

b. L80: How frequently were crabs fed?

c. L81: Was the 0.8g of oyster tissue weighed before or after the refrigeration period?

d. L83: How long before crab feeding was treatment water collected from the aquaria?

e. How did you collect control water? For it to be a valid control, then you must have had a similar aquarium setup and water exchange schedule as the 2 crab treatments. Please clarify.

f. What temperature was the water in oyster tanks and crab aquaria?
• Some more detail is also needed to replicate the crushing force measurements. For example, where on the shell was the force applied? How much area was the force applied to?
• Did you collect any measurements of the shells that were or were not crushed? Although the shell weights did not vary significantly among treatments, shells could have varied in some other size metric (i.e., the distribution of that weight across width, height, etc.).
• Related, differences in crushing force should be adjusted for variation in shell size. Otherwise, variation in crushing force could be due to variation in oyster size (length, width, height, some ratio of these), whether that variation is random or due to treatments.
• Please provide some reference or context for the crushing forces you are measuring, i.e., what is the crushing force of a blue crab claw? This is important for evaluating the ecological relevance of your results.

Validity of the findings

I have some questions/concerns about the data and statistical analyses.
a. First, I am not convinced that removing the random effect of “tank” from the full model is valid based on a small improvement in the AICc scores of the simpler model. Because of the experimental design, not including tank in the model leads to pseudoreplication. I don’t think including ‘tank’ will qualitatively change your results.

b. Related to (a): in Figure 1, what is the sample size for each SE? Because each tank is the independent replicate, these valued should be calculated with n = 13 (tank averages), rather than n = 65 oysters.

c. There are mesocosms missing from the data. Were these excluded from analyses? It seems that way based on the DFs. If so,
- please describe in methods and give final sample sizes for each treatment
- make sure your ANOVAs are set up to deal with an unbalanced experimental design. This is not necessarily straightforward in R.

d. Just a suggestion: why not just use simple linear contrasts to compare your treatments? With such a simple experiment and clearly described a priori hypotheses in the intro, I don’t see the need for overly conservative Tukey tests.

Additional comments

Overall, the paper is well-written. The experiment is simple, the results are clearly discussed. I would like to see more reference to the inducible defense literature, which is really what the paper is about.

The title is a bit confusing—I expected the predators and scavengers to be different species—and the phrase “induce stronger non-consumptive effects” doesn’t quite make sense. Maybe it’s the redundancy? The predator has a stronger nonconsumptive effect on oyster shell morphology or induces the prodiction of stronger shells suggest. I’d suggest something like “predatory blue crabs induce stronger shells in eastern oysters than scavenging blue crabs.”

Reviewer 3 ·

Basic reporting

In this study, the authors examined the effects of blue crabs fed aged oyster tissue and blue crabs fed fresh oyster tissue on oyster shell growth (strength and weight) responses to examine if prey can differentiate the risks associated with scavenging vs. live-feeding predators. Several studies have addressed if predators respond differentially to dead vs. live prey cues, but I am unaware of studies examining responses of prey to predators fed these diets, thus the question is a novel one. I think there are some moderate revisions needed prior to publication.

The structure and writing style are clear and properly reflect the structure of a professional journal article. It’s a short and simple study with an appropriate short and simple background.

Experimental design

The experimental design is appropriate for the study, but could use some clarifications in a few areas as well as different stats analysis. Specific comments on the experimental design and methods are:
- Please address if and how many oysters and crabs died during the study.
- How often were crabs fed?
- It is important in diet cue study designs to distinguish between the possibilities of prey responding specifically to different diet cues coming off the predator (i.e. in crab pee) vs. synergistic responses of sensing an injured conspecific + predator cue. It’s a bit hard to follow your water replacement and cue preparation to know if/how you addressed this possibility. (In general, I found lines 82-85 a bit hard to follow in terms of methodology).
- Why did you choose to use an oyster aged via refrigeration vs. aged in tank seawater.? Microbial growth is likely to influence the smell of the oyster and the cue response, yet you chose a methodology that slows microbial growth and is likely less representative of scavenge in natural environments. It isn’t a fatal flaw, but the reasoning should be addressed.
- Line 68 – oyster initial length- please include SE
- Line 92 – settings for the amplifier –do these numbers need units? What do these numbers correspond to for those of us unfamiliar with this instrumentation?
- Instead of a one way ANOVA, I suggest the authors re-analyze the data with a Repeated measures ANOVA. Using 5 oysters from a single treatment tank is not an independent measure (as the data was analyzed), but is actually a repeated non independent measure of the same treatment. I think the authors tried to address this problem using tank as a random factor, but a random blocking factor does not adequately address the non-independence of the oysters from each tank. The type of analysis is significant, especially given the marginally significant differences between some treatments.

Validity of the findings

The findings and conclusions of the article were consistent with the data shown. The authors should include the figure with shell weight, even though the data were not statistically significant.
The authors adequately address why they think the response in to the different cues may differ, but they do not discuss the different findings of shell strength vs. weight. In this case, shell strength supports their hypothesis, but shell weight doesn’t. This is likely because of different metrics of shell growth (thickness, length, etc), but the authors bypass what they hypothesize is causing the change in strength, especially given that weight of the shells doesn’t change. Please discuss your non-significant findings and how they likely relate to oyster growth patterns and relations to shell strength.
The authors should also address why they didn’t choose to examine other growth metrics, such as length or width. (I suspect it is likely because oysters grow in odd shapes making measurements difficult). One methodological suggestion to examine what growth metrics are changing (thickness, length etc) in response to cues would be to examine Surface Area:Weight ratios. SA could easily be measured with an imaging program. This could be done on this data set if the authors still have their dried shells to increase the findings of this study. Alternatively, just take it as a suggestion for future studies.

Additional comments

See other suggestions below:
Line 36-37: Predator exudates are more commonly used that other sensory cues in many aquatic environments, but not necessarily in terrestrial ones.
Line 49: Studies of dietary classification are rare. – I don’t agree with this statement as it relates to cue studies. There are many many studies that examine how predator diet influences prey response. Reword or clarify

---

## Round 0.2 · accepted · Accept

The authors have done an admirable job of addressing the comments and suggestions of the 3 reviewers.